# General features of the retinal connectome determine the computation of motion anticipation

**Jamie Johnston*†, Leon Lagnado**

School of Life Sciences, University of Sussex, Brighton, United Kingdom

**Abstract** Motion anticipation allows the visual system to compensate for the slow speed of phototransduction so that a moving object can be accurately located. This correction is already present in the signal that ganglion cells send from the retina but the biophysical mechanisms underlying this computation are not known. Here we demonstrate that motion anticipation is computed autonomously within the dendritic tree of each ganglion cell and relies on feedforward inhibition. The passive and non-linear interaction of excitatory and inhibitory synapses enables the somatic voltage to encode the actual position of a moving object instead of its delayed representation. General rather than specific features of the retinal connectome govern this computation: an excess of inhibitory inputs over excitatory, with both being randomly distributed, allows tracking of all directions of motion, while the average distance between inputs determines the object velocities that can be compensated for.

**\*For correspondence:**
J.Johnston@leeds.ac.uk

**Present address:** †School of Biomedical Sciences, University of Leeds, Leeds, United Kingdom

**Competing interests:** The authors declare that no competing interests exist.

## Introduction

The brain estimates the location of an object in visual space by reading out which retinal ganglion cells (RGCs) respond to it. This 'retinotopic map' is preserved through the visual pathway and is used to guide behaviour (*Hubel and Wiesel, 1962*; *Malpeli and Baker, 1975*; *Bonin et al., 2011*). There is, however, a problem to overcome when this map is used to estimate the position of a moving object: the slow speed with which photoreceptors convert light into an electrical signal causes RGCs to respond ~70 ms after an object first appears (*Baylor and Hodgkin, 1974*). During this delay, a tennis ball served by a professional player will have travelled ~3–4 m. The fact that the position of the ball can be estimated precisely enough to meet it with a racquet demonstrates that the visual system is able to overcome the phototransduction delay for moving stimuli. Together, these computations are termed motion anticipation, and they begin in the inner retina (*Berry et al., 1999*): when a stimulus is moving, the peak-firing rate of retinal ganglion cells (RGCs) occurs earlier than expected from the delayed response to a flashed stimulus. This correction supports accurate target tracking in salamander (*Leonardo and Meister, 2013*).

Although the retina is capable of processing the visual input in a variety of ways, we still do not understand how most of these transformations are achieved (*Olveczky et al., 2003*; *Münch et al., 2009*; *Gollisch and Meister, 2010*; *Bölinger and Gollisch, 2012*). One aspect of motion processing that is understood in detail is the generation of directionally selective responses peculiar to a small subset of RGCs (*Borst and Euler, 2011*). Analysis of neuronal connectivity using large-scale electron microscopy has demonstrated that this computation involves asymmetric connections with a specific type of inhibitory interneuron, the starburst amacrine cell (*Briggman et al., 2011*), which has dendrites that are themselves directionally selective (*Hausselt et al., 2007*; *Yonehara et al., 2013*). Motion anticipation appears to be a more fundamental retinal computation than directional selectivity because it is observed in the large majority of ganglion cells, of different functional types and

**eLife digest** The retina is a structure at the back of the eye that converts light into nerve impulses, which are then processed in the brain to produce the images that we see. It normally takes about one-tenth of a second for the retina to send a signal to the brain after an object first moves into view. This is about the same time it takes a tennis ball to travel several meters during a tennis match, yet we are still able to see where the moving tennis ball is in real time. This is because a process called 'motion anticipation' is able to compensate for the delay in processing the position of a moving object. However, it was not known precisely how motion anticipation occurs.

Inside the retina, cells called photoreceptors detect light and ultimately send signals (via some intermediate cell types) to nerve cells known as retinal ganglion cells. These signals can either excite a retinal ganglion cell to cause it to send an electrical signal to the brain, or inhibit it, which temporarily prevents electrical activity. Each cell receives signals from several photoreceptors, which each connect to a different site along branch-like structures called dendrites that project out of the retinal ganglion cells.

Johnston and Lagnado have now investigated how motion anticipation occurs in the retina by using electrical recordings of the activity in the retinas of goldfish combined with computer simulations of this activity. This revealed inhibitory signals, sent from photoreceptors to retinal ganglion cells via a type of intermediate cell (called amacrine cells), play a key role in motion anticipation. The ability to track motion effectively in all directions requires more inhibitory signals to be sent to the dendrites of a retinal ganglion cell than excitatory signals. These two types of input must also be randomly distributed across the cell. Furthermore, it is the density of these input sites on a dendrite that determines how well the retina can compensate for the motion of a fast-moving object. The building blocks required for motion anticipation in the retina are also found in visual areas higher in the brain. Therefore, further work may reveal that higher visual areas also use this mechanism to predict the future location of moving objects.

across many species (*Berry et al., 1999*; *Schwartz et al., 2007*). We do not understand how the retina generates motion anticipation, but the ubiquity of this process across ganglion cell types suggests that it reflects general properties of the inner retinal circuit rather than the specific wiring of subtypes of neuron. A fast decrease in the gain with which signals are transmitted through the retina has been proposed to account for motion anticipation, but the site(s) of such control have not been identified (*Berry et al., 1999*; *Schwartz et al., 2007*; *Chen et al., 2013*). One possibility is that moving stimuli induce a fast decrease in the efficiency of excitatory transmission from bipolar cells to ganglion cells, shortening the time-course of excitation. This idea is attractive because bipolar cells are the only route by which excitatory signals are transmitted to ganglion cells, and these synapses have been identified as a major site of gain-control, with an increase in temporal contrast causing fast depression of vesicle release (*Demb, 2008*; *Jarsky et al., 2011*; *Nikolaev et al., 2013*). A second possibility is that some mechanism intrinsic to ganglion cells alters the time-course of the ganglion cell response (*Chen et al., 2013*). We now need to test these various possibilities experimentally.

We have used electrophysiology and modelling to investigate the mechanisms of motion anticipation and find that it is not usually exerted through changes in the excitatory inputs to RGCs, but rather depends on the shunting effect of inhibition that RGCs receive from amacrine cells. The non-linear interaction between excitatory and inhibitory synapses is a result of the passive properties of the dendritic tree and remains intact when active conductances are blocked. Motion anticipation operates across most ganglion cells because it depends on general rather than specific features of the retinal connectome: independent and random distributions of excitatory and inhibitory synapses on RGCs (*Freed and Sterling, 1988*; *Hitchcock, 1989*; *Kolb and Nelson, 1993*; *Jakobs et al., 2008*; *Xu et al., 2008*; *Schwartz et al., 2012*); an excess of inhibitory inputs (*West, 1976*; *Koontz and Hendrickson, 1987*; *Freed and Sterling, 1988*; *Marshak et al., 1988*; *Hitchcock, 1989*; *Kolb and Nelson, 1993*; *Haverkamp et al., 1997*; *Zhu and Gibbins, 1997*; *Owczarzak and Pourcho, 1999*; *Marshak et al., 2002*; *Jakobs et al., 2008*; *Koizumi et al., 2011*), and an average

excitatory synaptic spacing along dendrites of ~5 µm (*Freed and Sterling, 1988*; *Jakobs et al., 2008*; *Xu et al., 2008*; *Koizumi et al., 2011*). The excess of inhibitory inputs allows tracking of all directions of motion, while the average distance between excitatory inputs determines the object velocities that can be compensated for. This study demonstrates how a computation fundamental to the retinal circuit can be understood in terms of general properties of the connectome and the cable properties of dendrites.

## Results

### Motion anticipation does not result from changes in the gain of excitatory transmission

The basic phenomenon of motion anticipation is demonstrated in *Figure 1*, where we recorded extracellular spikes from individual RGCs in retinal flat-mounts from goldfish. The receptive field (RF) of each ganglion cell was mapped using bars flashed in a random order across the retina (*Johnston et al., 2014*), and then a bar was flashed onto the centre of the receptive field: the delay to peak firing averaged 62 ± 3 ms, largely reflecting the delay in phototransduction (*Baylor and Hodgkin, 1974*) (n = 25; *Figure 1A,C*). But when the same bar (−100% contrast; 160 µm wide, equivalent to 2.4° of visual angle) was moved across the retina at 500 µm s$^{-1}$ (7.5° s$^{-1}$), the peak spike rate occurred 46 ± 13 ms (n = 25) *before* the leading edge reached the receptive field centre and then activity decayed *before* the bar left (*Figure 1B,C*). The time at which a RGC responded most strongly therefore encoded the anticipated position of the bar in retinotopic space (*Berry et al., 1999*) (*Figure 1B,C*). Motion anticipation was observed across many functional types of ganglion cell, including brisk-transient, brisk-sustained and orientation selective cells. RGCs with larger RFs tended to display greater anticipation, with RF size accounting for 35% of the observed variance in the delays for motion (Pearson's r = −0.593, n = 25, *Figure 1D*). The three cells that failed to show any motion anticipation also had the smallest RF size.

The timing of the peak spike response in RGCs might be brought forward if a moving stimulus caused excitation in the ganglion cell to be truncated soon after it began (*Berry et al., 1999*). Such a rapid decrease in the gain of the excitatory input might be caused by (i) depression intrinsic to the bipolar cell synapse, as occurs during contrast adaptation (*Rieke, 2001*; *Demb, 2008*; *Nikolaev et al., 2013*) and/or (ii) feedback inhibition, either reciprocal or lateral, that bipolar cell terminals receive from amacrine cells (*Roska et al., 2000*; *Tanaka and Tachibana, 2013*) (*Figure 1E*). Alternatively, truncation of the spike response might reflect feedforward inhibition from amacrine cells onto RGCs (*Figure 1F*). To differentiate between these possibilities, we began by isolating the excitatory postsynaptic current in ganglion cells and asking whether the excitatory input generated by a moving stimulus displayed any degree of motion anticipation.

Our standard moving stimulus, a bar 2.4° wide moving at 7.5° s$^{-1}$, spends 320 ms at any one point on the retina. When this bar was presented statically for 320 ms over the RF centre, the bipolar cell input decreased rapidly after a short delay (*Figure 1G*, n = 12). This decay reflects a combination of two mechanisms controlling the output from bipolar cells: feedback inhibition from amacrine cells (*Roska et al., 2000*; *Tanaka and Tachibana, 2013*) and depression intrinsic to the synaptic terminal, which reflects depletion of vesicles in a state ready for rapid fusion (*Rieke, 2001*; *Demb, 2008*; *Nikolaev et al., 2013*). To test whether these presynaptic mechanisms of gain control could generate motion anticipation in RGCs we measured the time-course of the EPSC in response to the moving bar. In 7 out of 9 RGCs the peak excitatory input was *delayed*, occurring 158 ± 34 ms after the bar reached the receptive field centre (*Figure 1H*, black trace), demonstrating that the fast gain reduction in the EPSC was not sufficient to generate motion anticipation. Further, the time-course of the EPSC induced by motion was not significantly different from the linear response predicted by convolving the dynamics of the static EPSC measured in *Figure 1F* with each ganglion cell's RF (Kolmogorov–Smirnov test, n = 7), indicating that there was no correction at all for the lag in phototransduction. These results rule out events at the bipolar cell terminal as a mechanism of motion anticipation, be they intrinsic depression, feedback inhibition or lateral inhibition (*Figure 1E*).

Two RGCs that we recorded from provided an interesting exception to this pattern: the motion-induced EPSC was significantly truncated over the latter half of the receptive field compared to the linear prediction (*Figure 1I*). Notably, these were the only two RGCs out of 25 that we sampled to

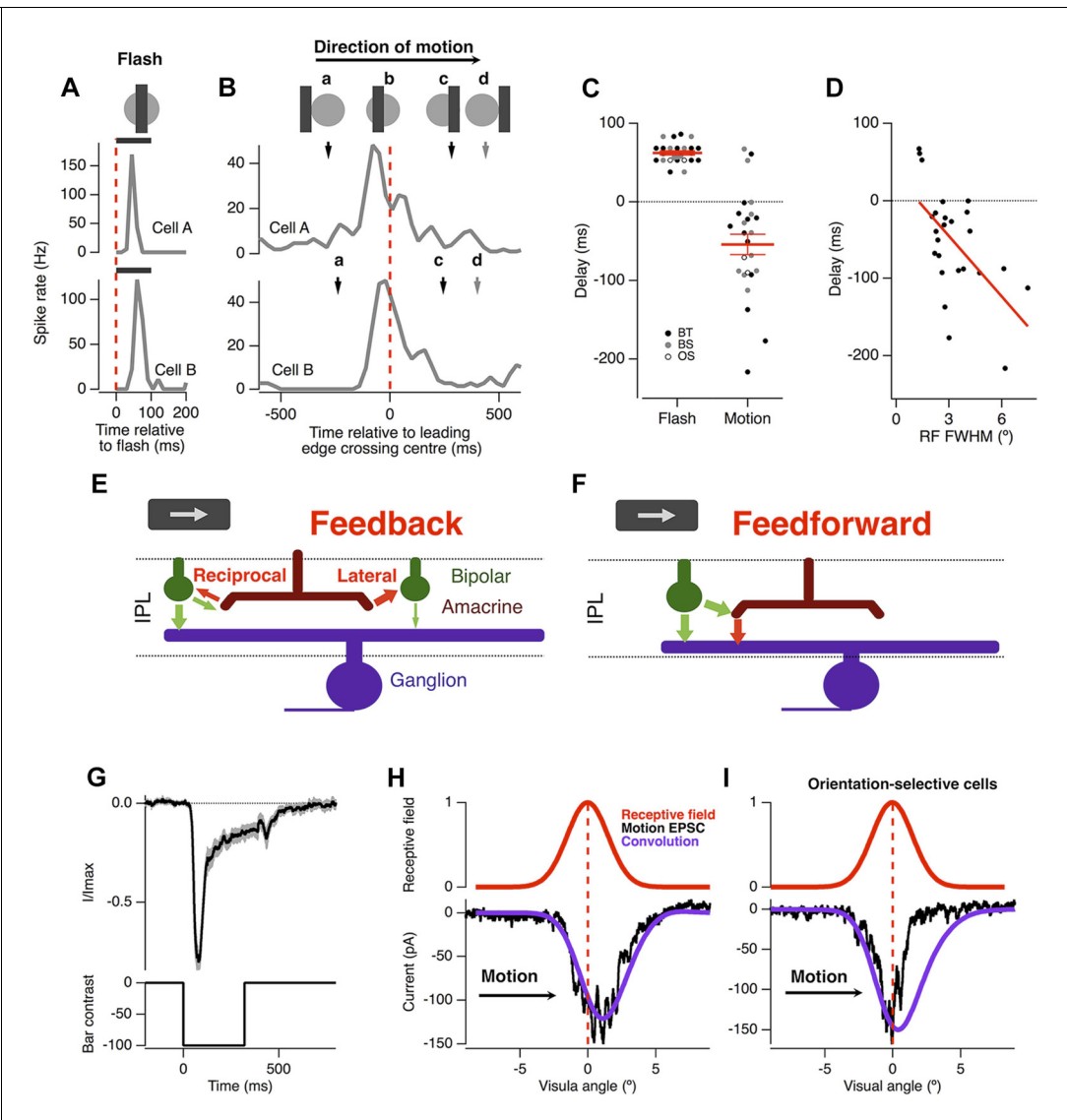

**Figure 1.** Motion anticipation in the retina is not due to a gain change in bipolar cells. (A) An example of two ganglion cells responding to a bar flashed on their receptive field centres for 100 ms (−100% contrast). (B) The response of the same cells to a bar of width 160 μm (2.4°) moving at 500 μm s$^{-1}$ (7.5° s$^{-1}$). The position of the bar relative to a cell's receptive field is shown above for the different time points indicated by the lettered arrows. (C) A comparison of the delay for the maximal response to a flashed stimulus (62 ± 2.6 ms) and the time of maximal spiking to a moving stimulus relative to the time at which the stimulus reached the centre of the RF (−46 ± 12.6 ms; n = 25 ganglion cells; p < 0.0001). BT = Brisk-transient, BS = Brisk Sustained, OS = Orientation-selective, see 'Materials and methods' for cell classification. (D) The degree of motion anticipation was correlated with the RF size (Pearson's r = −0.593, n = 25). (E) Schematic of retinal feedback circuits in the inner plexiform layer (IPL), excitation and inhibition are represented by green and red arrows respectively. (F) Schematic of feed-forward inhibition in the IPL. (G) The dynamics of the EPSC evoked by a −100% contrast bar flashed over the RGCs RF centre for 320 ms. Individual cells were normalised before averaging (n = 12, SEM in grey). (H) Example of the EPSC recorded as a bar moves across the receptive field of an OFF ganglion cell (average of six presentations). The peak EPSC lags behind the receptive field centre by 79 ± 17 μm (1.2 ± 0.3°, n = 7). The purple line indicates the expected linear response obtained from convolution of the receptive field with the EPSC in F. The motion evoked EPSC was not significantly different from the expected linear response, indicating that lateral inhibition is not present (using the Kolmogorov–Smirnov test, n = 7). (I) Orientation selective cells did show a clear indication of lateral inhibition (n = 2).

show orientation-selective responses. However, the large majority of RGCs in goldfish and other species are not orientation-selective (*Levick, 1967*), indicating that most signals transmitted by bipolar cells are not corrected for the lag in phototransduction.

## Feedforward inhibition is necessary for motion anticipation

The result in *Figure 1* immediately suggest that the inhibitory input which RGCs receive directly from amacrine cells (*Lukasiewicz and Werblin, 1990*; *Roska et al., 2000*; *Masland, 2012*; *Cafaro and Rieke, 2013*) may be responsible for motion anticipation. To test the role of feedforward inhibition we measured the motion response of individual RGCs before and after the selective disruption of their inhibitory inputs. *Figure 2A* shows a RGC's spiking in response to the standard moving stimulus, recorded in cell-attached mode. As usual, firing occurred just before the leading edge reached the receptive field centre (average of 18 ± 10 ms, n = 6). We then disrupted inhibition in this single ganglion cell by going into whole-cell mode and dialyzing the cell with an intracellular solution containing 120 mM Cl⁻ (*Figure 2A*, grey). The advantage of this approach over pharmacology or genetic manipulation is that it acutely disrupts the inhibition impinging on a *single* RGC while leaving excitatory inputs and the rest of the retinal circuitry intact; a similar approach was used to demonstrate direction selectivity occurs postsynaptically in direction-selective RGCs (*Taylor et al., 2000*). Disrupting inhibitory inputs greatly enhanced the response to a moving bar, indicating that under normal circumstances inactivation of $Na_v$ channel does not attenuate the RGC response to motion. Importantly, with inhibition disrupted, the location of the peak firing became delayed occurring 210 ± 26 ms after the stimuli had reached the RF centre (*Figure 2C*, p < 0.0002). In contrast, for the four cells tested, the delay for a flash was unaffected by disrupting inhibition (64.4 ± 8.7 ms vs 62.7 ± 5.2 ms, *Figure 2D*). We conclude that feedforward inhibition from amacrine cells to RGCs plays the major role in correcting for the lag in phototransduction allowing the retina to correctly signal the position of a moving object.

## The passive properties of dendrites are sufficient to account for motion anticipation

To investigate the biophysical basis of motion anticipation we constructed computational models of three RGCs whose morphologies were recovered with 2-photon microscopy (*Figure 3C*). Although many neurons contain active dendritic conductances (*Magee and Johnston, 1995*; *Bischofberger and Jonas, 1997*; *Hausselt et al., 2007*), including some RGCs (*Oesch et al., 2005*; *Sivyer and Williams, 2013*), we began by exploring the simpler situation in which excitatory and inhibitory conductances interact with just passive properties, as this is the backbone for electrical signaling in dendrites (*London and Häusser, 2005*). The time-course of synaptic conductance changes

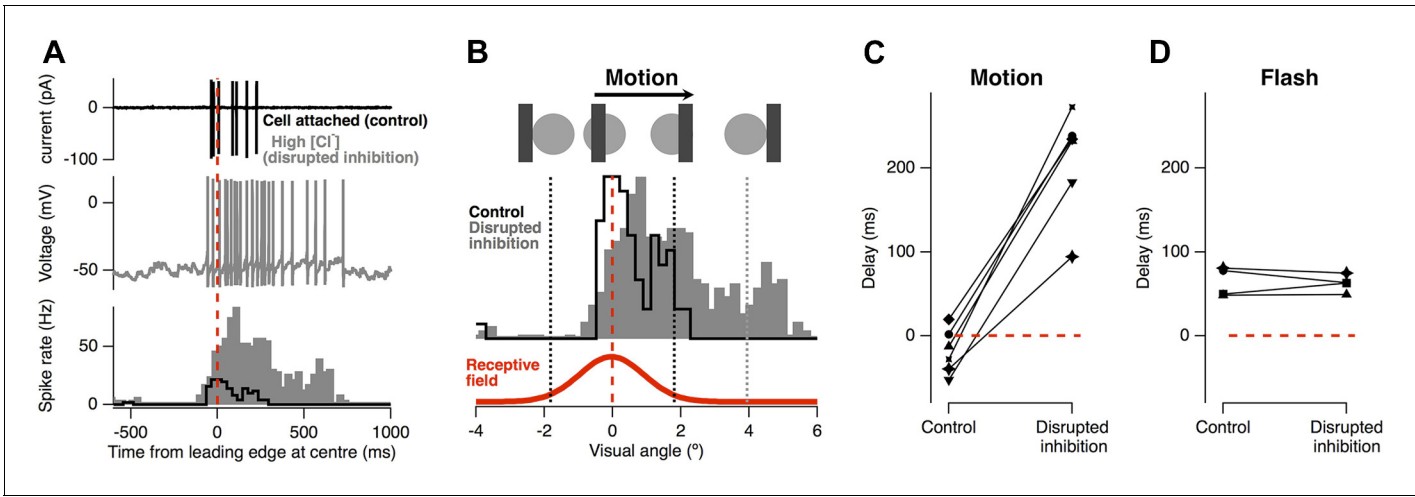

**Figure 2.** Feed-forward inhibition is necessary for motion anticipation. (**A**) Top: cell-attached recording from a single RGC as a 160 µm bar moves across the retina at 500 µm s⁻¹ (7.5° s⁻¹). Middle: whole-cell recording in the same cell 15 min after going whole-cell with 120 mM Cl⁻ in the pipette. Bottom: spike-time histograms calculated from 20 repetitions of the stimulus for each condition. (**B**) The normalised spike rates from a plotted as a function of the distance of the bars leading edge from the RF centre, which is shown below in red. (**C**) When inhibition was disrupted by introduction of high Cl⁻, the peak spike rate shifted from −18 ± 11 ms *before* the leading edge reached the centre to 209 ± 26 ms after the centre was traversed (n = 6; p = 0.0002). (**D**) The delay in response to a flash was not affected by disruption of inhibition (64.4 ± 8.7 ms vs 62.7 ± 5.2 ms, n = 4).

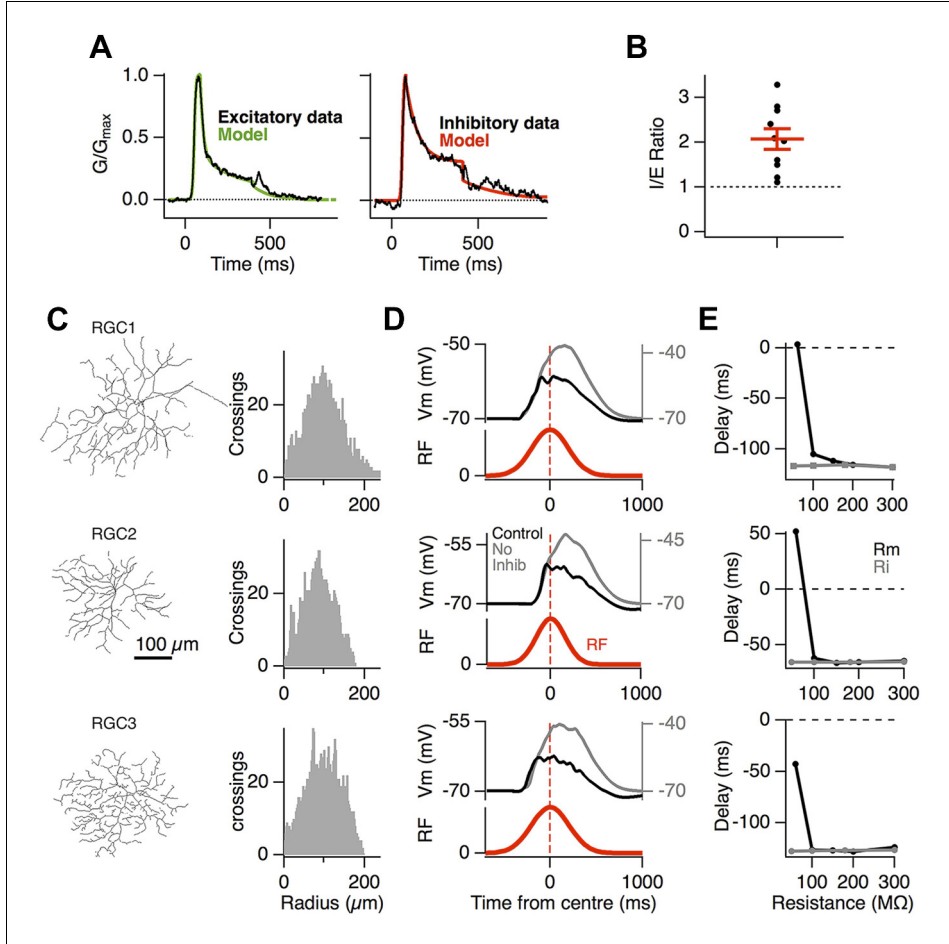

**Figure 3.** Motion anticipation in model ganglion cells depends on feedforward inhibition. (**A**) The time course of the synaptic inputs used in the model were constructed with piecewise functions fit to the synaptic conductance evoked from a 320 ms −100% contrast step over the RF centre, equivalent to the stimulus that the moving bar generates at any one point on the retina (average data shown in black and fits in colour; see 'Materials and methods'). (**B**) The inhibitory-to-excitatory conductance ratio measured over the RF centre in response to −100% contrast bar was 2.1 ± 0.23 (n = 10). (**C**) Line drawings of the 3 RGCs used for modeling, with Sholl plots to their right indicating the number of dendritic crossings for spheres of increasing distance from the soma. Excitatory synapses were placed randomly across the dendritic trees to give an average inter-synapse distance of 4.7 μm. Inhibitory synapses were also distributed randomly giving inhibitory to excitatory synapse ratios of 2.36:1, 2.04:1 and 2.41:1 for the three RGCs shown. All synapses had identical weights. (**D**) The output of each model RGC in response to a 160 μm bar moving across its dendritic field at 500 μm s$^{-1}$, with (black) and without inhibition (grey), the corresponding RF is shown below in red. (**E**) Motion anticipation was robust to changes in the membrane resistance (Rm) and axial resistance (Ri).

The following figure supplement is available for figure 3:

**Figure supplement 1.** The voltage-clamped EPSCs (black) in response to the same moving stimuli as *Figure 3D*, plotted relative to the RF (red).

used in simulations mimicked those measured experimentally (*Figure 3A*), and the average density of synaptic inputs over the dendritic trees also matched known distributions with around one excitatory synapse per 5 μm of dendrite (*Freed and Sterling, 1988*; *Jakobs et al., 2008*; *Xu et al., 2008*) and a ~2:1 ratio of excitatory to inhibitory synapses, reflecting the ratio of the inhibitory to excitatory conductance measured over the receptive field centre (*Figure 3B*, 2.1 ± 0.23, n = 10). Electron microscopy has consistently demonstrated that inhibitory synapses outnumber excitatory

inputs in RGCs and the 2:1 ratio we used for simulations is relatively conservative; inhibition-to-excitation (I/E) ratios up to 5:1 have been observed in some ganglion cell types (*Freed and Sterling, 1988*; *Hitchcock, 1989*; *Kolb and Nelson, 1993*; *Owczarzak and Pourcho, 1999*). These synapses were distributed randomly over the dendritic tree, with the total numbers of 2960, 3450 and 3120 in the three reconstructed cells that we used. Full details of the simulations are provided in 'Materials and methods'.

When the receptive field of these model neurons was probed by mimicking flashed bars, it reproduced a Gaussian receptive field similar to that measured experimentally (*Figures 1*, *3D*). Crucially, the EPSPs arriving at the soma displayed motion anticipation when tested with our standard moving bar stimulus (*Figure 3D*), demonstrating that the passive interactions of excitatory and inhibitory synaptic inputs are sufficient to generate motion anticipation. The model also predicted the effects observed experimentally after disrupting inhibition: when inhibitory synapses were removed, the peak EPSP was delayed, occurring 163 ms *after* the bar entered the receptive field. These basic features of motion anticipation were observed in all of the modelled RGCs, despite their different branching patterns (*Figure 3C,D*). Consistent with our observations in *Figure 1*, the voltage-clamped excitatory currents in these model ganglion cells failed to display motion anticipation (*Figure 3—figure supplement 1*).

The integration of voltages in dendrites is dependent on the membrane resistance (Rm) and intracellular resistance (Ri) (*Rall, 1957*), and in the RGCs that we recorded from, Rm varied from 83 MΩ to 580 MΩ, with a median of 218 MΩ (n = 37). The ability of feedforward inhibition to generate motion anticipation was robust to variations in both Rm and Ri over this range, but was destroyed once Rm fell to 50 MΩ (*Figure 3E*). These simulations therefore indicate that the passive cable properties of RGCs can account for motion anticipation across different dendritic morphologies and electrical properties of RGCs.

## Active dendritic conductances are not necessary for motion anticipation

Voltage-sensitive channels in dendrites can modulate integration of synaptic inputs; for example, NMDA receptors boost EPSPs that are activated in a temporal order moving towards the soma (*Branco et al., 2010*) and a proximal-to-distal gradient of voltage-sensitive $Ca^{2+}$ channels in the dendrites of starburst amacrine cells acts to boost EPSPs moving from the soma toward distal dendrites (*Hausselt et al., 2007*). To test directly whether active conductances were required to generate motion anticipation, we recorded from ganglion cells whose dendrites were made passive by dialysis of 2 mM MK-801 and 10 mM QX-314-Bromide; together these, substances block NMDA receptors and voltage-sensitive $Na^+$ and $Ca^{2+}$ channels (*Talbot and Sayer, 1996*; *Kuzmiski et al., 2010*). In addition to preventing any dendritic boosting of EPSPs, the somatic $Na_v$ channels were also blocked allowing us to observe the generator potential responsible for spiking. About 10 min after achieving whole-cell access we verified that the drugs had blocked voltage-gated channels by delivering a depolarizing current injection; by this time spikes were blocked and the resultant depolarization was well fit by a single exponential (*Figure 4A*). In separate experiments we found that 10 min was sufficient to completely fill the dendritic tree of ganglion cells with Alexa 488, which has a molecular weight around twice that of MK-801 and QX-314. We provided our standard moving stimulus to these passive cells and compared the motion-evoked EPSP to the expected linear response, obtained by convolution of the flash response with the measured RF (*Figure 4B*). All cells had EPSPs that peaked significantly earlier than the expected linear response (p = 0.0002, n = 7), occurring 47 ± 29 ms before the stimuli reached the RF centre and this was not significantly different to the time of peak spike rate observed for the cells in *Figure 1* (47 ± 29 ms, n = 7 vs 46 ± 13 ms, n = 25, p = 0.9844). These results indicate that active conductances are not necessary for the computation of motion anticipation in the dendrites of RGCs.

The ability of the retina to correct for the delay in phototransduction breaks down at high velocities (*Berry et al., 1999*). We plotted the location of maximal spiking relative to the receptive field centre as a function of velocity for 26 ganglion cells recorded on a multi-electrode array and found that motion anticipation occurred when the bar moved at velocities between 3.8° s$^{-1}$ (250 μm s$^{-1}$) and 14.7° s$^{-1}$ (1000 μm s$^{-1}$) but not at 27.8° s$^{-1}$ (2000 μm s$^{-1}$) or higher (*Figure 5A*). This behaviour was closely reproduced by the model (*Figure 5A*), which also predicted breakdown at velocities higher than ~14.7° s$^{-1}$ (1000 μm s$^{-1}$, *Figure 5B*). The velocity at which motion anticipation decreased is similar to that observed in salamander for similar sized objects (*Leonardo and Meister,*

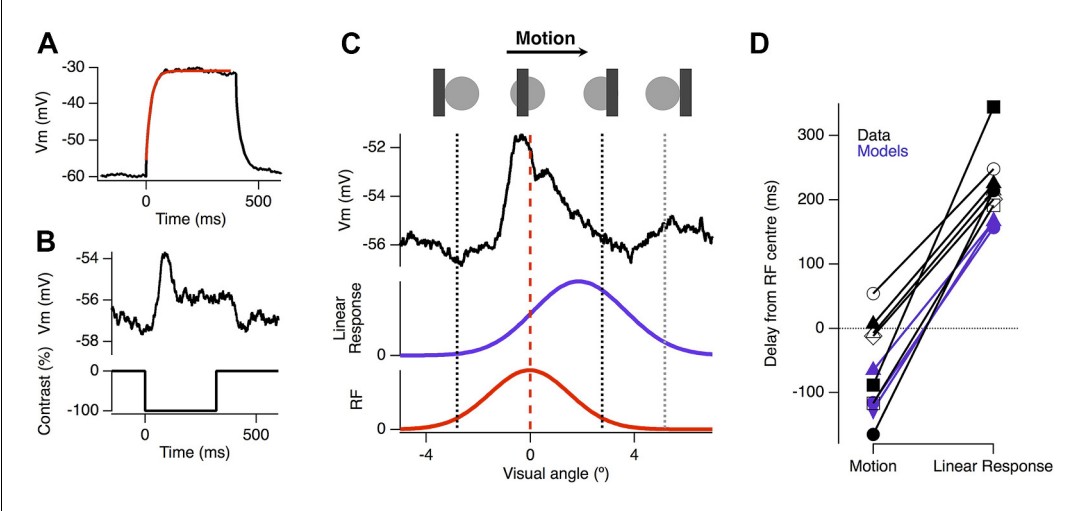

**Figure 4.** Motion anticipation is evident in the EPSPs of ganglion cells with passive dendrites. (**A**) Ganglion cells were dialyzed with 2 mM MK801 and 10 mM QX-314-Bromide. The efficacy of these drugs was assessed by attempting to fit a large step depolarization with a single exponential; the depolarization was well fit after 10 min of dialysis. (**B**) The EPSP evoked by a 160 μm (2.4°) bar flashed for 320 ms over the RF centre of the same ganglion cell in a with passive dendrites. (**C**) Example of the EPSP recorded as a bar moves across the receptive field of an OFF ganglion cell (average of 10 presentations). The EPSP is plotted as a function of the bar's location within the RF and the purple trace below, shows the expected linear response obtained by convolution of the EPSP from **B** with the RF. (**D**) The average delay for EPSPs in RGCs with passive dendrites was −47.2 ± 29.4 ms, whereas the expected linear response was always delayed with an average of 233.84 ± 19.6 ms (n = 7, p = 0.0002). The delays for the three models are shown in purple, with the linear response representing the model with only excitation.

*2013*). Passive integration of synaptic inputs is therefore sufficient to provide a quantitative account of several fundamental features of motion anticipation.

## A biophysical explanation of motion anticipation

Motion anticipation causes a ganglion cell to spike preferentially when an object enters its receptive field, and we have shown that feedforward inhibition is necessary for this phenomenon (*Figures 2–5*). But why does inhibition have a stronger effect over the latter half of the receptive field? To investigate the biophysical mechanisms underlying motion anticipation in more detail, we explored a model RGC with a radially symmetric dendritic tree. *Figure 6A* plots the membrane potential across a dendrite spanning the receptive field as a bar moves across, and compares this to the voltage experienced by the soma. In the absence of inhibition, the somatic voltage was an accurate reflection of the depolarization seen in the dendrites (*Figure 6A*, left). However, with inhibition present, this was only the case for the first half of the RF, once the stimulus crossed the RF centre dendritic depolarizations had less influence on the somatic voltage. As a result, the peak excitatory drive occurred close to when the bar traversed the RF centre.

Why does the latter half of the dendritic field appear to electrically uncouple from the soma? In a landmark theoretical study Koch et al. laid out the conditions that make inhibitory synaptic inputs most effective at counteracting excitatory inputs: (a) inhibitory inputs should be located on the path between the excitatory synapse and the soma (i.e., proximally along the dendrite), and (b) inhibitory inputs should be activated *before* the distal excitatory synapse (*Koch et al., 1983*; *Liu, 2004*; *Hao et al., 2009*; *Pouille et al., 2013*). Satisfying these conditions depends critically on whether the object is moving towards the soma or away from it. This idea is explained further in *Figure 6B* where we examined the influence of a single inhibitory synapse on the ability of a single excitatory synapse to depolarize the soma. As the moving stimulus enters the receptive field an inhibitory input distal to the soma is activated first (*Figure 6B*, left), but it fails to attenuate depolarization because it is not on the path between the excitatory synapse and the soma (condition a). If the inhibitory input is located proximal to the excitatory input, it will still be ineffective because it is activated *after* the excitatory drive has reached the soma (*Figure 6B*, middle; condition b). Only when the stimulus is traversing the latter half of the RF does proximal inhibition occur just *before* distal excitation to 'cut-

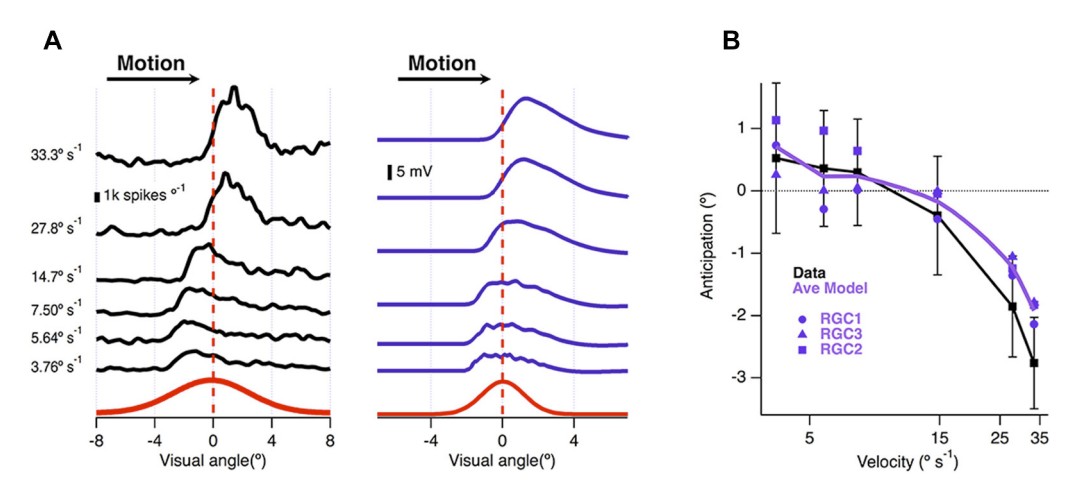

**Figure 5.** Velocity-dependence of motion anticipation. (**A**) Black traces show the spike histograms from a single cell for a 160 μm bar moving at different velocities plotted against the position of the leading edge relative to the RF centre (average of 30 presentations). Purple traces show the average response of the model (RGC2) to the same stimulus parameters, with an I/E ratio of 2.04:1. Note that the peak EPSP starts to lag behind the receptive field centre at higher velocities. (**B**) The average amount of anticipation plotted against velocity for 26 OFF ganglion cells (black, ±SD). Motion anticipation operated until a velocity of about 1 mm s$^{-1}$, as also predicted by the three models (purple).

out' excitation reaching the soma (*Figure 6B*, right). In this way, the peak-firing rate is brought 'forward' in time to compensate for the delay inherent in phototransduction.

## Motion anticipation depends on a general feature of the retinal connectome

The conditions in which proximal inhibition blocks excitation arriving at the soma that were laid out by *Koch et al., 1983* immediately suggest a wiring rule for the generation of motion anticipation: place inhibitory synapses in positions proximal to nearby excitatory synapses. Motion anticipation operates for objects moving across the receptive field in any direction, so this pattern would be expected across the whole dendritic tree. Such an organization might be achieved if developmental processes cause inhibitory synapses to be placed systematically in positions proximal to the nearest excitatory synapse. Such a specific wiring rule does not, however, fit with anatomical studies, which consistently indicate that excitatory and inhibitory inputs are distributed independently and at uniform density along RGC dendrites (*Freed and Sterling, 1988*; *Hitchcock, 1989*; *Kolb and Nelson, 1993*; *Owczarzak and Pourcho, 1999*; *Jakobs et al., 2008*; *Xu et al., 2008*; *Schwartz et al., 2012*). These two constraints, proximal inhibition across the whole dendritic tree, together with random positioning of synapses, would be satisfied if there are more inhibitory inputs per dendrite than excitatory ones, and this is observed: anatomical studies consistently demonstrate that inhibitory synapses outnumber their excitatory counterparts in the large majority of RGCs (*West, 1976*; *Koontz and Hendrickson, 1987*; *Freed and Sterling, 1988*; *Marshak et al., 1988*; *Hitchcock, 1989*; *Kolb and Nelson, 1993*; *Haverkamp et al., 1997*; *Zhu and Gibbins, 1997*; *Owczarzak and Pourcho, 1999*; *Marshak et al., 2002*; *Jakobs et al., 2008*; *Koizumi et al., 2011*).

To investigate how the ratio of inhibition to excitation affected the generation of motion anticipation we carried out simulations in which we varied the number of inhibitory inputs in the model cells. When the ratio of inhibition to excitation (I/E) was increased, the resultant EPSPs became smaller (*Figure 7A*) and the peak of the EPSP shifted forward in time (*Figure 7B*). In two of the model ganglion cells, the temporal shift in the time-course of excitation began with I/E > 1, while in the third neuron it began with I/E > 2 (*Figure 7C*). These observations confirm that the generation of motion anticipation requires an excess of inhibitory inputs when synaptic sites are positioned randomly and independently over the dendritic tree.

The ability of the retinal circuit to extrapolate motion breaks down at higher object velocities, measured as greater than 1 mm s$^{-1}$ in most RGCs in salamander, rabbit and goldfish (*Berry et al.,*

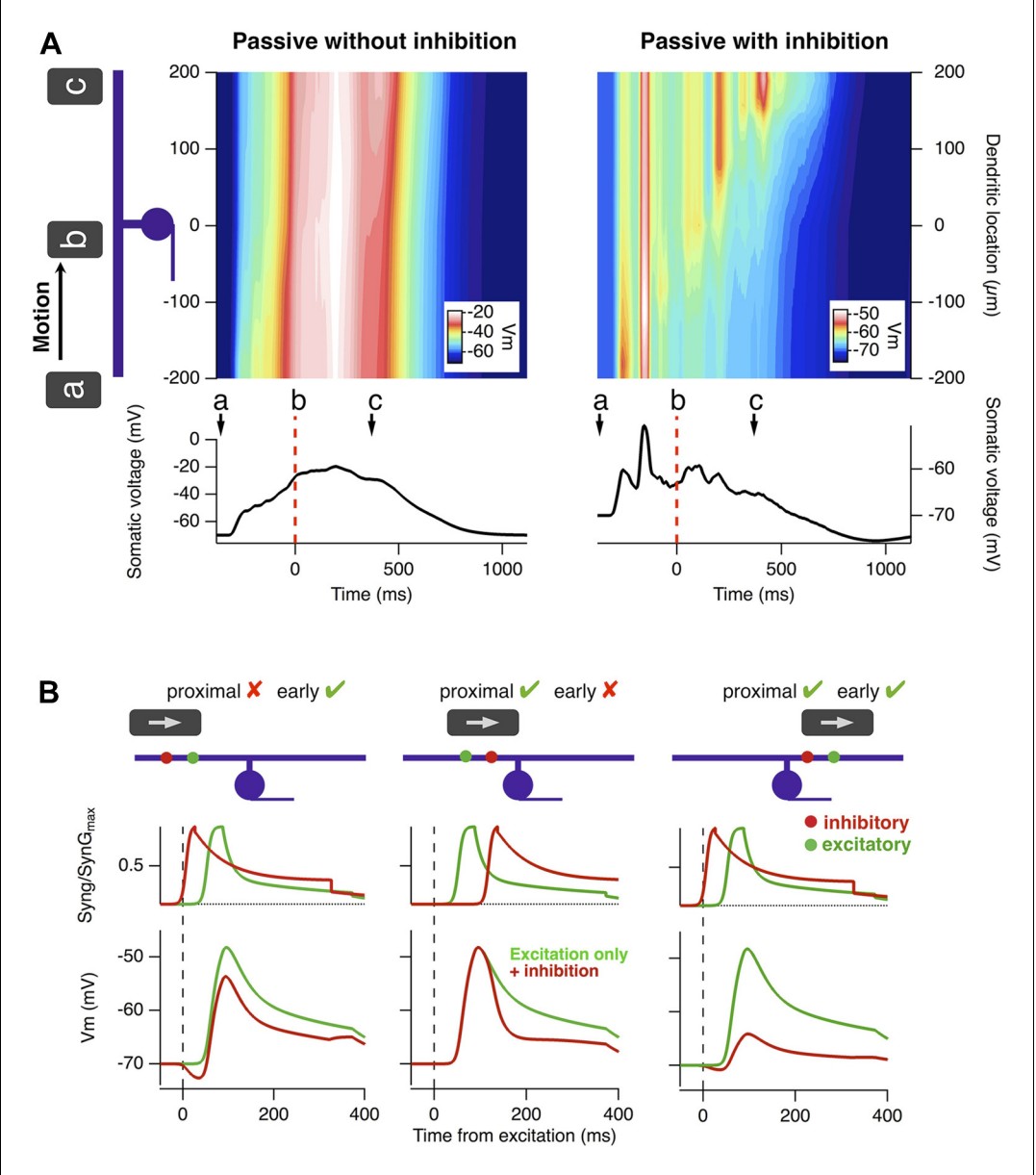

**Figure 6.** A biophysical explanation of motion anticipation. (**A**) Heat plots of the voltage across the dendritic tree of a simplified model RGC as a function of time in response to a moving bar. The voltage at the soma is plotted below. Left: with only excitation present the somatic voltage follows the voltage seen in the dendrites. Right: with inhibition present, the somatic voltage only followed dendritic excitation in the first half of the RF, then hyperpolarized as the bar moved across the remainder. Note: the arrow marked a represents the stimulus entering the RF; b is time zero, when the leading edge reaches the RF centre, and c marks the leading edge reaching the distal edge of the dendritic field. (**B**) The positions of inhibitory synapses (red) relative to excitatory synapses (green) strongly affects the depolarisation observed at the soma. Left: For a stimulus moving across the initial half of the RF, distal inhibition is activated first attenuating depolarisation of the soma only modestly. The red voltage trace is the somatic response with inhibition present and the green trace is with only excitation. Middle: Proximal inhibition in the initial half of the receptive field has little effect on spiking as these synapses are activated later than the distal excitation. Right: For a stimulus moving across the distal half of the RF, proximal inhibition is activated before distal excitation and is very effective at reducing depolarisation of the soma by more distal excitatory synapses.

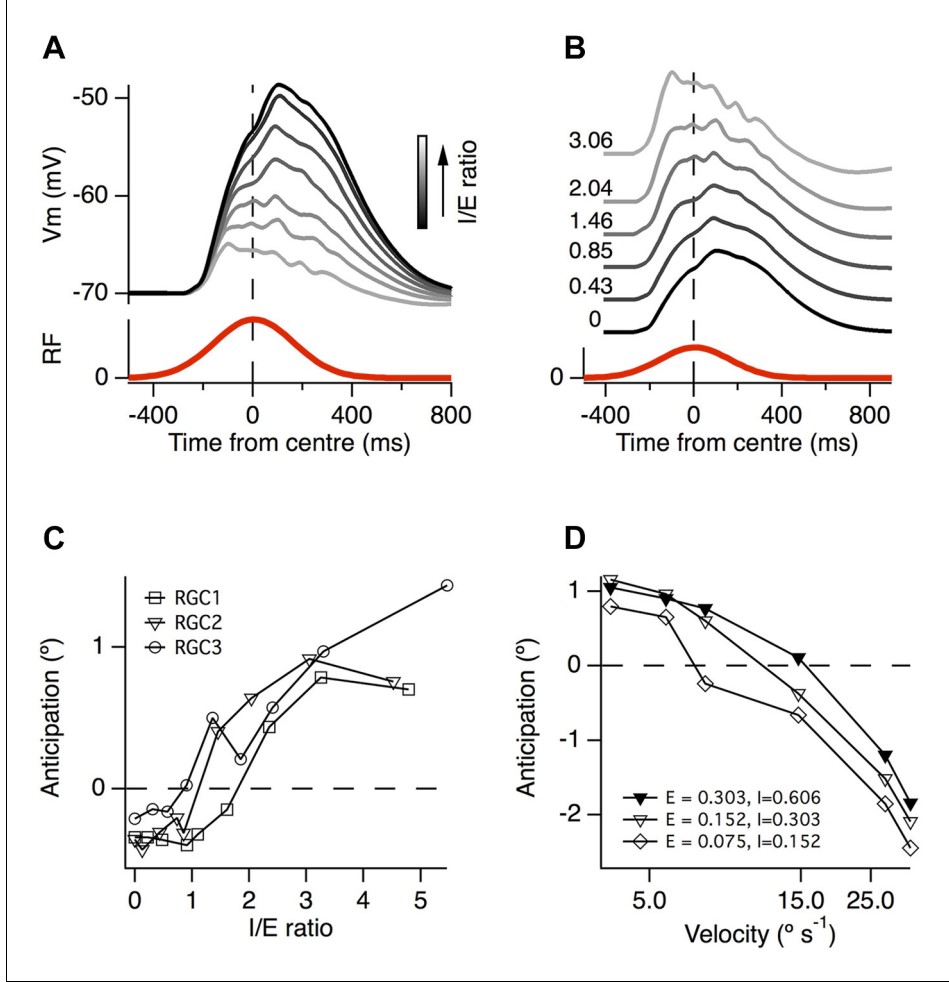

**Figure 7.** A greater ratio of inhibition to excitation is important for motion anticipation. (**A**) The output of a model RGC in response to the standard moving stimulus for various ratios of inhibitory to excitatory synapses. The number of excitatory synapses was fixed and the darkest trace represents 0 inhibitory synapses. As expected the amplitude of the response becomes smaller with increased inhibition. (**B**) The amplitudes of the EPSPs in a, normalised to compare the time course of the EPSPs. Note that as inhibition increases the peak of the EPSP moves forward in time, no longer occurring with a delay. (**C**) The amount of motion anticipation plotted as a function of the inhibition to excitation ratio (I/E). Note that motion anticipation emerges as the inhibitory synapses start to outnumber the excitatory. (**D**) The velocity dependence of motion anticipation was influenced by the synaptic density. The density of synapses was varied for RGC2 while keeping the I/E ratio fixed. Densities shown for excitatory (E) and inhibitory (I) as synapses $\mu m^{-1}$.

*1999*; *Figure 5B*). What aspect of the circuit determines this fundamental characteristic? Our simulations demonstrated that a key variable was the average distance between excitatory and inhibitory inputs. Several authors have noted that the distance between excitatory synapses is surprisingly consistent over the dendritic tree of most ganglion cells and across many species, at ~1 synapse per 5 $\mu m$ (*Freed and Sterling, 1988*; *Jakobs et al., 2008*; *Xu et al., 2008*; *Koizumi et al., 2011*). Together with a 2:1 I/E ratio, this characteristic distance predicted the observed critical velocity of ~1 mm $s^{-1}$ in our simulations (*Figure 5B*). When the synaptic density was reduced to a quarter of its initial value, while maintaining a constant I/E ratio, the critical velocity at which motion anticipation broke down decreased from ≥1 mm $s^{-1}$ to ~0.5 mm $s^{-1}$ (*Figure 7D*). It therefore appears that the density of synaptic inputs impinging on a RGC's dendritic tree determines the critical velocity at which motion extrapolation breaks down.

## Discussion

This study demonstrates how the retina compensates for the slow speed of phototransduction so that the neural image transmitted to the brain conveys information about the present position of a moving object rather than its position ~70 ms in the recent past. The fundamental observation is that motion anticipation arises from the passive interactions of excitatory and inhibitory inputs within the dendritic tree and three general properties of the retinal connectome: the excess of inhibitory over excitatory inputs in RGC dendrites, the average density of these inputs and their random distribution over the whole dendritic tree.

### Generality of the mechanism

Our measurements in goldfish indicate that the amount of motion anticipation is correlated with the RF size, with small RF exhibiting the least anticipation (*Figure 1D*). In primate and monkey the specialized midget ganglion cell pathway has RF smaller than those presented here, consequently they may not exhibit motion anticipation. Indeed, measurements of synapse ratios obtained from electronmicroscopy of parafoveal RGCs in monkey and human indicate that some of these small RGCs have I/E ratios of ~1:1 (*Kolb and Dekorver, 1991*; *Calkins et al., 1994*; *Calkins and Sterling, 2007*), which would further diminish their ability to perform motion anticipation. This observation may make sense in the framework of separate 'what' and 'where' visual pathways (*Goodale and Milner, 1992*); ganglion cells with the smallest RFs, such as the midget or parvocellular (P) pathway, are involved in coding object detail and correspondingly have the highest spatial resolution but slowest conduction velocities (*Gouras, 1969*). Whereas RGCs with larger RFs, the magnocellular (M) pathway, have high conduction velocities (*Gouras, 1969*) and are involved in coding the spatial location of an object relative to the organism, a computation that would obviously benefit from motion anticipation. For the RGCs involved in the 'where' pathway, three general properties of the retinal connectome that give rise to motion anticipation are observed across a wide variety of species: an excess of inhibitory over excitatory inputs, a constant density of these inputs and their random distribution over the whole dendritic tree (*West, 1976*; *Koontz and Hendrickson, 1987*; *Freed and Sterling, 1988*; *Marshak et al., 1988*; *Hitchcock, 1989*; *Kolb and Nelson, 1993*; *Weber and Stanford, 1994*; *Haverkamp et al., 1997*; *Zhu and Gibbins, 1997*; *Owczarzak and Pourcho, 1999*; *Marshak et al., 2002*; *Jakobs et al., 2008*; *Koizumi et al., 2011*; *Schwartz et al., 2012*). The generality of these conditions across species underlines the fundamental importance of correcting for the phototransduction delay so that retinotopic mapping can be conserved when encoding the position of a moving object. This study therefore provides an example of the way in which the general rules of retinal wiring determine a key computation of visual processing.

### Motion anticipation is autonomous to each ganglion cell

We find that the excitatory signal transmitted to RGCs is delayed relative to the retinotopic position of a moving object, indicating that the temporal correction occurs within the ganglion cell (*Figure 1H*). A priori reasoning also suggests that motion anticipation should be autonomous to each ganglion cell, rather than occurring earlier in the retinal circuitry. Consider a single bipolar cell situated between, and contacting two ganglion cells (*Asari and Meister, 2012*). For a stimulus traversing the retina this bipolar cell will drive one ganglion cell at its latter edge and the second ganglion cell at its initial edge. If the gain of this bipolar cell were reduced for the first ganglion cell it would also retard the second cell's ability to respond to the same stimulus. In effect the neural image on the retina would fade as the stimulus moved across visual space. Instead, motion anticipation is generated by passive interactions within the ganglion cell dendritic tree, and is therefore relatively independent of other computations carried out by the circuitry of the inner retina. It is the non-linear interaction of inhibitory and excitatory synaptic inputs within the dendritic tree that shifts the profile of excitation forward in time. Simulations indicate that the density of synaptic inputs along dendrites determines the object velocities that can be corrected.

Dendrites are a fundamental computational unit in the nervous system. In many brain regions they perform non-linear operations on the inputs they receive (*Hausselt et al., 2007*; *Branco and Häusser, 2010*; *Smith et al., 2013*), these can result simply from their passive properties or through active membrane conductances (*London and Häusser, 2005*). The modelling we carried out as part of this study illustrates why the classical linear-nonlinear receptive field model incompletely describes

the response of a ganglion cell to a moving stimulus: excitatory and inhibitory synaptic inputs interact in a non-linear manner that depends on time and position on the dendritic tree. *Koch et al., 1983* have described the shunting effect of an inhibitory input lying on the path between an excitatory input and the soma as an analogue implementation of a Boolean AND-NOT operation in which one input vetoes another. Here we find that this AND-NOT operation is activated when the object moves away from the soma and receptive field centre, aligning the time of peak excitation closer to the time when the object is in the centre of the receptive field.

Dendrites of retinal ganglion cells have also been shown to contribute to other non-linear transformations of the visual input. For example the local integration of excitatory and inhibitory inputs in PV-5 cells endows them with sensitivity to approaching objects (*Münch et al., 2009*), whereas in directionally selective ganglion cells dendritic Na$_v$ channels are used to boost their direction sensitivity (*Schachter et al., 2010*; *Sivyer and Williams, 2013*). We show that motion anticipation emerges from the passive properties of dendrites, but it is possible that subtypes of RGCs could augment anticipation using active conductances. For example, NMDA receptors selectively boost EPSPs that move centripetally along a dendrite (*Branco et al., 2010*), such a mechanism could augment motion anticipation in guinea pig OFF α RGCs where NMDA receptors make a significant contribution to visually evoked spiking but not in OFF ∂ or ON α RGCs where NMDA receptors are less conspicuous (*Manookin et al., 2010*). Additionally dendritic spikes combined with gap junction coupling allow a specific subtype of directionally selective ganglion cell to respond much earlier than expected to motion in its preferred direction (*Trenholm et al., 2013*).

## Inhibitory wiring in the retina

A fundamental finding of our study is that motion anticipation in ganglion cells arises when feedforward inhibitory synapses outnumber excitatory inputs (*Figure 7C*). We modelled each inhibitory synapse as purely feedforward (*Figure 1F*), a ubiquitous and simple circuit motif that numerous types of amacrine cell provide for example, A2 narrow-field, A8 bistratified, A13 and A22 amacrine cells (*Kolb, 2005*). There are, however, at least 22 different types of amacrine cell (*MacNeil and Masland, 1998*), but we understand little about the specific response properties of these and even less about how the different types are connected to ganglion cell dendrites. The few notable exceptions highlight the sophistication that can be achieved. For example a combination of intrinsic (*Hausselt et al., 2007*) and synaptic (*Lee and Zhou, 2006*) mechanisms endow individual dendrites of starburst amacrine cells with a directional preference for moving stimuli. Our model demonstrates (*Figure 6B*) that the location of an inhibitory synapse on the dendritic tree of a ganglion cell can have a large effect on the output of that ganglion cell. Indeed the operation of direction-selective ganglion cells results from the particular wiring of their starburst amacrine cell inputs (*Briggman et al., 2011*). It is therefore expected that a proportion of the inhibitory inputs impinging on ganglion cells may already represent complex transformations of the visual signal.

A concerted effort to elucidate the functional roles of amacrine cells in the retina is now required. Genetically-encoded indicators can be used to image the synaptic output of neurons in the retina (*Dreosti et al., 2009*; *Odermatt et al., 2012*) and targeting these reporters to different subtypes of amacrine cells should reveal their functional diversity. High-resolution connectomics (*Briggman et al., 2011*; *Helmstaedter et al., 2013*) is now uncovering how amacrine cells wire to the dendritic trees of different ganglion cells. The challenge in the future is to marry these two approaches so that the functional landscape can be overlaid on the connectomic map to guide explorers of the retina.

## Inhibition-to-excitation ratios

The balance between excitation and inhibition profoundly affects the gain and tuning of neural responses (*Isaacson and Scanziani, 2011*). For instance, in the transition from the anesthetized to the awake state, increased inhibition within the visual cortex is correlated with pyramidal cell responses that are briefer in time and more narrowly tuned in space (*Haider et al., 2013*). It has long been recognized that non-midget RGCs receive many more inhibitory inputs than excitatory and that these are located randomly over the dendritic tree, and independently of excitatory inputs (*Freed and Sterling, 1988*; *Hitchcock, 1989*; *Kolb and Nelson, 1993*; *Weber and Stanford, 1994*; *Owczarzak and Pourcho, 1999*). This characteristic wiring can now be understood in the context of

the mechanism generating motion anticipation; a surplus of inhibitory inputs on each dendrite ensures that stimuli entering a receptive field are transmitted more effectively to the soma than stimuli traversing the latter half.

Our simulations (*Figure 7*) and the analysis of *Koch et al., 1983*; demonstrate that variations in the distance between inhibitory and excitatory inputs will vary the object velocity that provides the most effective shunting of excitation to the soma because this shunting depends on the time delay between these conductances. The velocity tuning of motion anticipation is therefore expected to vary within individual dendrites. Nonetheless, the average intersynaptic spacing observed over many RGCs is surprisingly constant (*Jakobs et al., 2008*; *Koizumi et al., 2011*) and reproduces the velocity tuning observed physiologically (*Figure 5B* and *Figure 7D*).

It appears that the general conditions that generate motion anticipation in the retina, topographically organized neurons receiving feedforward inhibition, also exist in downstream areas of the visual pathway, including the superior colliculus or optic tectum (*Bollmann and Engert, 2009*), thalamus (*Blitz and Regehr, 2005*) and area V1 of the visual cortex (*Miller, 2003*; *Haider et al., 2013*). A topographic organization of sensory inputs combined with feedforward inhibition onto dendrites may therefore be a general mechanism for correcting time delays in neural signals relative to events in the external world.

## Materials and methods

### Electrophysiology

All procedures were carried out in accordance with the UK Animals (Scientific Procedures) act 1986. Retinae were removed from goldfish (15–20 cm long) and placed in AMES solution (Sigma–Aldrich, Gillingham, UK) diluted to 270 mOsM. Pieces of retina ~1 cm$^{-2}$ were mounted photoreceptor side down in a recording chamber and perfused at 2–3 ml min$^{-1}$ with AMES bubbled with 95% $CO_2$/5% $0_2$. Retinal ganglion cells were visualized under infrared light using a camera and recordings made with an Axopatch 200B (Molecular Devices, Sunnyvale, CA). Extracellular recordings were made in voltage-clamp mode. For whole-cell voltage-clamp recordings, the intracellular solution contained 104 mM $CsMeSO_4$, 8 mM CsCl, 5 mM $Na_2$ Phosphocreatine, 4 mM HEPES, 2 mM Mg.ATP, 1 mM Na.GTP, 1 mM EGTA and 2 mM QX-314-Chloride, with this solution the calculated reversal potential for Cl$^-$ was −59 mV. Voltage-clamping at −60 mV and 0 mV isolated the EPSC and IPSC respectively. For whole-cell current-clamp measurements with high Cl$^-$ the intracellular solution contained 120 mM KCl, 10 mM $Na_2$ Phosphocreatine, 4 mM HEPES, 2 mM Mg.ATP, 1 mM Na.GTP, 0.15 mM EGTA. For the passive dendrite recordings in *Figure 4* the intracellular solution contained 104 mM $KMeSO_4$, 8 mM KCl, 10 mM $Na_2$ Phosphocreatine, 10 mM QX-314-Bromide, 4 mM HEPES, 2 mM MK801, 2 mM Mg.ATP, 1 mM Na.GTP, 0.15 mM EGTA. Pipettes had a resistance of 5–6 MΩ. Signals were digitized using an ITC18 A-D converter and acquired on a Mac mini using Neuromatic running in Igor Pro 6 (Wavemetrics, Lake Oswego, OR).

### Multi-electrode array recordings

Spikes were recorded from RGCs on a 60 channel multi-electrode array (Multichannel Systems, Reutlingen, Germany) using perforated electrode arrays. Spikes were sorted as described previously (*Johnston et al., 2014*) using Wave_clus (*Quiroga et al., 2004*). Stimuli were repeated 30 times each. Only cells with responses to the standard moving bar stimulus at all velocities were included for further analysis.

### Visual stimulation

A 852 × 600 pixel monochromatic OLED micro-display (eMagin, part number EMA-100100, Bellevue, WA) was focused onto the photoreceptor layer of the retina through an oil condenser. Pixels measured 4 × 4 μm on the retinal surface. Visual stimuli were delivered via Matlab (Mathworks, Natick, MA) using psychophysics toolbox libraries. Visual stimulation and electrophysiology were synchronized by recording the times of screen refreshes and the timing precision was verified with PMTs. The mean irradiance was 40 nW mm$^{-2}$ and our standard stimulus was a bar of −100% contrast measuring 160 μm by 2400 μm on the retina. To relate the retinal images to objects in the real world we measured the distance between the centre of the lens and the retina, for the goldfish used

in this study (~150 mm in length) this was ~3.8 mm. All sizes and speeds have been converted to degrees of visual angle according to the equation:

R/n = tanV,

where R is the size of the retinal image, n is the distance from the retina to the lens centre and V is visual angle. Therefore our standard stimulus of a 160 μm bar covered ~2.4° of visual angle and moved at ~7.5° s$^{-1}$.

## Cell classification

Initially RGCs were classified as OFF vs ON and brisk-transient vs brisk-sustained by flashing full field stimuli of −100 and 100% contrast for 0.5 s each. Brisk transient cells responded strongly to stimulus onset then adapted completely. Brisk-sustained cells responded strongly initially and then adapted slightly over the rest of the stimuli. No ON cells were encountered in our recordings. We also tested for both direction-selectivity and orientation-selectivity by moving a bar across the retina at four different angles. We encountered two orientation-selective cells but no direction-selective cells. The orientation selective cells responded strongly to a moving bar in their preferred orientation and failed to spike to a bar orthogonal to this axis.

## Measuring RGC receptive fields

Similar to (*Johnston et al., 2014*), the RF of each RGC was mapped by flashing a −100% contrast 80 μm (1.2°) bar at pseudo-random locations across a single axis of the retina (the same used for motion stimuli). The order was then deshuffled and the total spikes for each bar was counted in a 150 ms window starting at each flash time, a Gaussian was then fit to this data.

## Digitizing RGC morphologies

RGCs were filled with 50 μM alexa 488 by dialysis through the patch pipette then, subsequent to electrophysiological recording, a volume containing the RGC dendrites and soma was acquired with a custom built 2-photon microscope similar to (*Esposti et al., 2013*). The morphologies of each RGC were then digitized using the 'Simple Neurite Tracer' plugin for ImageJ; these were then exported as SWC format, down sampled using custom written scripts (available at http://www.igor-exchange.com/project/DendritePruner) and imported to neuroConstruct.

## Models

Morphologically realistic models of RGCs with synaptic inputs were constructed in neuroConstruct (*Gleeson et al., 2007*) and ran in NEURON (*Hines and Carnevale, 2001*). The electrical properties used throughout the manuscript were: intracellular resistance (Ri) = 180 Ω.cm, membrane capacitance (Cm) = 1 μF cm$^{-2}$ and membrane resistance (Rm) = 20 MΩ cm$^{-2}$, however, we found that these parameters were not critical for the appearance of motion anticipation (*Figure 4*). The dendrites of the retinal ganglion cells traversed a virtual inner plexiform layer that was populated with excitatory bipolar and inhibitory amacrine synapses using the 'cubic close packed cell packing adaptor' in neuroConstruct. Bipolar terminals were packed with an effective radius of 5.1 μm and amacrine cells with an effective radius of 4 μm. We set connections between synapses and the dendritic tree of each RGC using morphology based connections. Searches for a connection between each synaptic terminal and a target dendrite were completely random with a distance constraint of 10 μm, 300 attempts were initiated for each synapse. This gave a density of 0.182, 0.303 and 0.189 excitatory synaptic contacts μm$^{-1}$ of dendrite for RGC1, 2 and 3 respectively, which is close to the observed density (*Freed and Sterling, 1988*; *Jakobs et al., 2008*; *Xu et al., 2008*). The density of inhibitory synapses was 0.426, 0.606 and 0.459 synapses μm$^{-1}$ of dendrite for each RGC, this higher density of inhibitory contacts is a conservative estimate of the number of inhibitory contacts seen with electron microscopy which can be as high as five times the number of excitatory contacts (*Freed and Sterling, 1988*; *Hitchcock, 1989*; *Kolb and Nelson, 1993*; *Owczarzak and Pourcho, 1999*). Each synapse was modeled as a point process, and the time-course of the currents were described by three piecewise functions obtained by fits to the measured synaptic conductances (*Figure 3A*). The three piecewise functions correspond to the onset of the steady state phase (a), the decay after the stimulus (b) and the adaptation to the stimulus over the steady state phase (c).

The synaptic conductance was G(t) = m × (a + b − c) where m is a scaling factor. For the excitatory conductance:

$$a(t) = \begin{cases} \frac{0.987}{1+e^{\frac{(52.9+t_0-t)}{5}}} & if\ t_0<t<t_0+t_d+52.9 \\ 0 & otherwise \end{cases},$$

$$b(t) = \begin{cases} 0.18*e^{\frac{(t_0+t_d-t)}{95.9}} & if\ t_0+t_d+52.9<t<t_0+t_d+475 \\ 0 & otherwise \end{cases},$$

$$c(t) = \begin{cases} 0.921+(-0.623*e^{\frac{(t_0+88-t)}{17.3}})+(-0.274+e^{\frac{t_0+88-t}{235.6}}) & if\ t_0+88<t<t_0+t_d+52.9 \\ 0 & otherwise \end{cases},$$

and for the inhibitory conductance:

$$a(t) = \begin{cases} \frac{1}{1+e^{\frac{(t_0+62.37-t)}{4.5}}} & if\ t_0<t<t_0+t_d+62.4 \\ 0 & otherwise \end{cases},$$

$$b(t) = \begin{cases} 0.2046*e^{\frac{(t_0+t_d-t)}{235.6}} & if\ t_0+t_d+1200<t<t_0+t_d+62.4 \\ 0 & otherwise \end{cases},$$

$$c(t) = \begin{cases} 0.7+(-0.63*e^{\frac{(t_0+82-t)}{69}}) & if\ t_0+82<t<t_0+t_d+62.4 \\ 0 & otherwise \end{cases},$$

where $t_0$ is the stimulus onset time and $t_d$ is the duration of the stimulus in ms. The delays inherent to the retinal circuitry, comprising both phototransduction delay and synaptic delay, are accounted for in these functions. Scaling factor m was set to 0.00003 ns for all synapses.

To facilitate assignment of an activation time ($t_0$) to each synapse, the array of bipolar and amacrine synapses were divided in to strips with a width of 30 μm. The RF of each model ganglion cell was measured by stimulating a single 30 μm strip and plotting the amplitude of the somatic EPSP vs space, similar to the performed physiological measurements. To simulate a bar moving across the retina each strip was given a set of values for $t_0$ that reflected the time required for the leading edge of the bar to traverse the 30 μm strip, each synapse was then randomly assigned a value from this set. For example, for a bar moving at 0.5 mm s$^{-1}$ the $t_0$ values of the first strip would range from 0 ms to 60 ms and for the subsequent strip range from 60 ms to 120 ms. For the simple model used in *Figure 6*, the soma had a diameter of 15 μm the primary dendrite extended for 20 μm then branched into eight radially symmetric dendrites of 200 μm length with a diameter of 1 μm.

## Analysis

All electrophysiogical data was analysed in Igor Pro. For the moving bars in *Figures 1*, *2*, spike times were detected by threshold crossings and histograms with 30 ms bins were then constructed from all presentations of the stimuli. For the analysis in *Figure 5* the data were binned in space at 15 μm bins for all velocities. The peak-firing rate and peak depolarization was determined using edge statistics with a threshold of 10%. that is, when the rate comes within 10% of the max that x value is used for the peak. For voltage-clamp each trial was presented at least six times and the average current from these presentations was used.

## Statistics

For comparison between the delays for a flash and motion in *Figure 1C* a paired t-test was used, all data were normally distributed. For the correlation in *Figure 1D* the Pearson's correlation coefficient was computed by bootstrapping with 100,000 samples. We used the Kolmogorov–Smirnov test to compare the measured EPSC in response to movement with that predicted by convolution of the EPSC resulting from a 320 ms flash and the receptive field (*Figure 1H,I*). A paired t-test was used to compare the peak firing before and after disrupted inhibition in *Figure 2C* and the passive EPSP with the linear response in *Figure 4*, data were normally distributed.

## Additional information

### Funding

| Funder | Grant reference number | Author |
| --- | --- | --- |
| Biotechnology and Biological Sciences Research Council | BB/L021528/1 | Leon Lagnado |

The funders had no role in study design, data collection and interpretation, or the decision to submit the work for publication.

### Author contributions

JJ, Conception and design, Acquisition of data, Analysis and interpretation of data, Drafting or revising the article; LL, Analysis and interpretation of data, Drafting or revising the article

### Ethics

Animal experimentation: All procedures were carried out in accordance with the UK Animals (Scientific Procedures) act 1986 and approved by the local ethical review committee at the University of Sussex.

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
