## [Decision Letter]

Thank you for sending your work entitled “General features of the retinal connectome determine the computation of motion anticipation” for consideration at *eLife*. Your article has been favorably evaluated by Eve Marder (Senior editor), Matteo Carandini (Reviewing editor), and by two reviewers, one of whom, Michael Berry, has agreed to share his identity.

The Reviewing editor and the reviewers discussed their comments before we reached this decision, and the Reviewing editor has assembled the following comments to help you prepare a revised submission.

In this excellent study, Johnston and Lagnado tackle an important and interesting phenomenon exhibited by signals in many types of retinal ganglion cells (RGCs): motion anticipation. This is the observation that peak RGC spiking activity (or that of a population of RGCs) is reached prior to a moving object arriving at the center of the receptive field. This observation is more than a decade old now, but the mechanisms that generate it are poorly understood.

To elucidate these mechanisms, the authors record from RGCs in the goldfish retina, and show these cells, like RGCs in several other species, exhibit motion anticipation. They then use a combination of electrophysiological, pharmacological, and computational/modeling approaches to show that motion anticipation results from a very generic mechanism: shunting inhibition on the RGC dendrites. They further show that dendritic active conductances and excitation (with feedback inhibition to bipolar cells) alone are not sufficient to give rise to motion anticipation.

The authors use models of 3 RGCs where they have measured dendritic architecture, along with synaptic densities taken from literature, to show that a 2:1 ratio of randomly placed inhibitory to excitatory synapses can account for the motion anticipation they observed in their recorded cells. Motion anticipation, therefore, can result from randomly distributed synapses as long as inhibitory synapses outnumber excitatory by about 2-to-1. Such conditions are met on many RGC dendrites (although see observation below) and might well be observed in other parts of the brain, making this biophysical mechanism potentially quite general. More generally, this study makes a compelling case that feedforward inhibition plays a major role in motion anticipation.

Also, the paper goes on to add some intuition underlying the phenomenon: strong shunting that occurs when inhibitory synapses are activated just before excitatory synapses located distally (as predicted a while ago by Koch et al.). In particular, Figure 6 is a very nice demonstration of the basic biophysical effect.

Overall, the manuscript makes a convincing case that feedforward inhibition is a central mechanism contributing to motion anticipation. This is a significant advance. What is less clear is whether it is the only mechanism. Below are some suggestions for improving the paper on this and other fronts. It should be possible to address these suggestions by editing the paper to highlight and discuss some of these issues, or perhaps by analyzing additional existing data, without necessarily requiring new experiments.

1) The manuscript claims that a general feature of the retinal connectome is 2:1 ratio in the number of inhibitory to excitatory synapses. The literature, however, does not seem to substantiate this claim. For instance the recent paper by Abbott et al., (2012) from the laboratories of Grunert and Martin working on the marmoset retina. They show that parasol and midget RGCs across a wide range of eccentricities have a 1:1 ratio of inhibition to excitation. Several studies from old world monkeys by Calkins, Sterling, Marshak, and Kolb indicate that inhibitory synapses in the peripheral retina outnumber excitatory synapses, but that the opposite is true in the central retina. Studies of bistratified RGCs in the monkey suggest that the proportion of excitatory and inhibitory synapses depends on dendritic strata. Thus the ratio of inhibitory to excitatory synapses depends on species, ganglion cell type, and retinal location.

Given that a 2:1 ratio does not appear to be a general feature, do the RGCs probed in this study exhibit a 2:1 ratio in synapse number? This seems important to determine to validate their analysis and conclusions. The authors show that the ratio of inhibition to excitation functionally is about 2:1, but doesn't synapse position/number also matter? Also, the authors show a large range of I:E ratios in different RGCs. Does the ratio of I:E correlate with the amount of motion anticipation, with low ratios exhibiting less anticipation? This seems straightforward to test on a cell by cell basis and would help to relate inhibition directly motion anticipation rather than by the process of elimination.

2) In Figure 1, signals are compared between a static bar and a moving bar. In Figure 2 they are only shown for a moving bar without inhibition. Does removing inhibition also postpone the time to a peak firing in response to a static bar? Showing this would clarify that we are looking at an interaction between a moving stimulus and inhibition, not just that inhibition shortens the response time-to-peak for any stimulus.

3) It would be nice to see the degree of motion anticipation of individual RGCs before and after dialyzing the cells with MK801 and QX-314-Bromide to eliminate active conductances. The authors show active conductances are not required, but it seems like they may still augment or amplify the amount of motion anticipation. As it stands the reader is left with the impression that active conductances don't participate, but it is not clear that this is the case. This question speaks to the issue of sufficiency. Currently, Figure 4 compares the amount of motion anticipation after drug wash-in to a linear model of the cell. A possibly stronger test would be to compare the amount of motion anticipation before and after drug wash in and show that it changes minimally.

4) How do the speeds of motion used in this study relate to speeds in the real world? The authors motivate their study by the important contribution the retina makes to the behavioral phenomenon of motion anticipation. However, without knowing the size of a goldfish eye, one cannot relate any of the speeds they probed experimentally to the speed of objects in the environment. The findings in this study could be more relevant to a general readership if this relationship is made explicit and speeds are quoted in units of degrees of visual angle per second. Related, it seems important to note that this will depend on the size of the animal's eye. Thus, given an equivalent amount of motion anticipation per RGC in a small eye compared to a large eye, there will be very different amounts of motion anticipation at the level of the retina. This seems worth noting in the Discussion.

---

## [Author Response]

1) The manuscript claims that a general feature of the retinal connectome is 2:1 ratio in the number of inhibitory to excitatory synapses. The literature, however, does not seem to substantiate this claim. For instance the recent paper by Abbott et al., (2012) from the laboratories of Grunert and Martin working on the marmoset retina. They show that parasol and midget RGCs across a wide range of eccentricities have a 1:1 ratio of inhibition to excitation. Several studies from old world monkeys by Calkins, Sterling, Marshak, and Kolb indicate that inhibitory synapses in the peripheral retina outnumber excitatory synapses, but that the opposite is true in the central retina. Studies of bistratified RGCs in the monkey suggest that the proportion of excitatory and inhibitory synapses depends on dendritic strata. Thus the ratio of inhibitory to excitatory synapses depends on species, ganglion cell type, and retinal location.

Given that a 2:1 ratio does not appear to be a general feature, do the RGCs probed in this study exhibit a 2:1 ratio in synapse number? This seems important to determine to validate their analysis and conclusions. The authors show that the ratio of inhibition to excitation functionally is about 2:1, but doesn't synapse position/number also matter? Also, the authors show a large range of I:E ratios in different RGCs. Does the ratio of I:E correlate with the amount of motion anticipation, with low ratios exhibiting less anticipation? This seems straightforward to test on a cell by cell basis and would help to relate inhibition directly motion anticipation rather than by the process of elimination.

The literature concerning the ratio of inhibitory to excitatory synapses is broad and consistently indicates that for non-midget ganglion cells, inhibitory synapses outnumber their excitatory counter parts. We state that this is a general feature as it has been detailed across many species, including cat (; ; ; ), goldfish (; Hitchcock 1989a), primates (including human) (; Jacoby et al. 1996; ), toad (), turtle () as well as the ground squirrel ().

The paper by Abbott et al. (2012) uses light microscopy and finds lower densities of inhibitory synapses than other studies using electron microscopy. The deficiencies in light compared to electron microscopy are well known. A point which the authors concede in their discussion: “In summary, the absolute density of the presumed amacrine input analyzed with our light microscopic methods is likely to be lower than absolute density estimated by electron microscopy.”

Furthermore, the method used to identify inhibitory synapses in this paper is likely to systematically exclude some inhibitory synapses, the authors state: “Thus, amacrine synapses containing the a1 subunit of the GABAA receptor are probably not included in counts of gephyrin IR puncta and thus also contribute to underestimation of amacrine input.”

The synapse positions used in our modeling reflect the known distributions of synapses; excitatory synapses are uniformly distributed across the dendrites and inhibition is distributed independently of these. This observation is also consistent across many species (; ; ; ; ; ; ; ).

We have included a new section in the Discussion on the generality of the mechanism and discuss this in light of the additional data we include showing that RF size is related to the amount of motion anticipation. We make specific reference to the lower I/E ratios observed in the paraforeal midget RGCs of primates.

For your reference, in the Results section, we added the following sentences:

“RGCs with larger RFs tended to display greater anticipation, with RF size accounting for 35% of the observed variance in the delays for motion (Pearson’s r = -0.593, n=25, Figure 1). The 3 cells that failed to show any motion anticipation also had the smallest RF size.”

In the Discussion, we included a new subsection entitled “Generality of the mechanism”: “Our measurements in goldfish indicate that the amount of motion anticipation is correlated with the RF size, with small RF exhibiting the least anticipation (Figure 1)[…]. This study therefore provides an example of the way in which the general rules of retinal wiring determine a key computation of visual processing.”

2) In Figure 1, signals are compared between a static bar and a moving bar. In Figure 2 they are only shown for a moving bar without inhibition. Does removing inhibition also postpone the time to a peak firing in response to a static bar? Showing this would clarify that we are looking at an interaction between a moving stimulus and inhibition, not just that inhibition shortens the response time-to-peak for any stimulus.

This was a nice suggestion and we thank the editor/reviewers for it. In four of the cells we also presented a flashed bar after disrupting inhibition. There was no change in the delay to peak firing with disruption of inhibition (control = 64.4 ± 8.7 ms vs disrupted inhibition = 62.7 ± 5.2 ms). This data has been added to a panel in Figure 2, and the text has been updated, in the subsection “Feedforward inhibition is necessary for motion anticipation”, to say:

“Importantly, with inhibition disrupted, the location of the peak firing became delayed occurring 210 ± 26 ms after the stimuli had reached the RF centre (Figure 2, p < 0.0002). In contrast, for the four cells tested, the delay for a flash was unaffected by disrupting inhibition (64.4 ± 8.7 ms vs 62.7 ± 5.2 ms, Figure 2).”

3) It would be nice to see the degree of motion anticipation of individual RGCs before and after dialyzing the cells with MK801 and QX-314-Bromide to eliminate active conductances. The authors show active conductances are not required, but it seems like they may still augment or amplify the amount of motion anticipation. As it stands the reader is left with the impression that active conductances don't participate, but it is not clear that this is the case. This question speaks to the issue of sufficiency. Currently, Figure 4 compares the amount of motion anticipation after drug wash-in to a linear model of the cell. A possibly stronger test would be to compare the amount of motion anticipation before and after drug wash in and show that it changes minimally.

Unfortunately, in this last set of experiments, we did not record motion-evoked spikes in the same cells prior to breaking into whole-cell mode. We had already demonstrated that the majority of RGCs display motion anticipation (Figure 1). Our modeling had predicted that integration of excitation and inhibition together with passive cable properties of dendrites was able to produce motion anticipation (Figure 3). Due to the wide applicability of this mechanism throughout the brain we thought it important to directly test whether the EPSPs in passive ganglion cells display motion anticipation. This was the primary goal of the experiments in Figure 4.

However, we have now included a comparison of the time of the peak EPSP in passive dendrites with that observed for the time of peak spike rate from the data in Figure 1, in the subsection “Active dendritic conductances are not necessary for motion anticipation”: “All cells had EPSPs that peaked significantly earlier than the expected linear response (P=0.0002, n=7), occurring 47 ± 29 ms before the stimuli reached the RF center and this was not significantly different to the time of peak spike rate observed for the cells in Figure 1 (47 ± 29 ms, n = 7 vs 46 ±13 ms, n=25, P= 0.9844).”

This paper focuses on how motion anticipation arises from a mechanism common to the majority of ganglion cells. We have not addressed whether active conductances augment motion anticipation; if this occurs it is likely to vary in a cell type specific manner. We have added our thoughts on this to the Discussion (subsection “Motion anticipation is autonomous to each ganglion cell”): “We show that motion anticipation emerges from the passive properties of dendrites, but it is possible that subtypes of RGCs could augment anticipation using active conductances. […] Additionally dendritic spikes combined with gap junction coupling allow a specific subtype of directionally selective ganglion cell to respond much earlier than expected to motion in its preferred direction ().”

4) How do the speeds of motion used in this study relate to speeds in the real world? The authors motivate their study by the important contribution the retina makes to the behavioral phenomenon of motion anticipation. However, without knowing the size of a goldfish eye, one cannot relate any of the speeds they probed experimentally to the speed of objects in the environment. The findings in this study could be more relevant to a general readership if this relationship is made explicit and speeds are quoted in units of degrees of visual angle per second. Related, it seems important to note that this will depend on the size of the animal's eye. Thus, given an equivalent amount of motion anticipation per RGC in a small eye compared to a large eye, there will be very different amounts of motion anticipation at the level of the retina. This seems worth noting in the Discussion.

For the goldfish sizes used in this study (∼150 mm in length), the distance between the lens centre and retina was ∼3.8 mm, this was estimated by freezing and bisecting eyes from 3 fish. Our standard stimuli of a 160 µm bar moving at 500 µm s^-1^ corresponds to a bar covering 2.4° of visual angle moving at 7.5° s^-1^. We have now added this calculation to the Methods section and updated all appropriate units to degrees of visual angle.